# MELCAP: A UNIFIED SINGLE-CODEBOOK NEURAL CODEC FOR HIGH-FIDELITY AUDIO COMPRESSION

## ABSTRACT

Neural audio codecs have recently emerged as powerful tools for high-quality and low-bitrate audio compression, leveraging deep generative models to learn latent representations of audio signals. However, existing approaches either rely on a single quantizer that only processes speech tasks, or on multiple quantizers that are not well suited for downstream tasks. To address this issue, we propose MelCap, a high-fidelity neural codec with a single codebook. By decomposing audio reconstruction into two stages, our method preserves more acoustic details than previous single-codebook approaches, while achieving performance comparable to mainstream multi-codebook methods. In the first stage, audio is transformed into mel-spectrograms, which are compressed in the image domain and quantized into compact single tokens using a 2D tokenizer. A perceptual loss is further applied to mitigate the over-smoothing artifacts observed in spectrogram reconstruction. In the second stage, a Vocoder recovers waveforms from the mel discrete tokens in a single forward pass, enabling real-time decoding. Both objective and subjective evaluations demonstrate that MelCap achieves quality on comparable to state-of-the-art multi-codebook codecs, while retaining the computational simplicity of a single-codebook design, thereby providing an effective representation for downstream tasks. Demos are available at here[1].

## 1 INTRODUCTION

Discrete audio tokens generated by neural audio codecs compress continuous audio signals into a compact discrete space while preserving perceptual quality and semantic content Mousavi et al. (2025), enabling reduced storage requirements and faster transmission than continuous embeddings Theis et al. (2017). These tokens serve as an efficient and flexible interface for downstream tasks such as Automatic Speech Recognition (ASR) Radford et al. (2022) Hsu et al. (2021), Text-To-Speech generation (TTS) Peng et al. (2024) Du et al. (2024), music generation Yang et al. (2024), and so on. Audio codecs typically consists of an encoder-quantizer-decoder structure to encode, where the encoder transforms the input waveform into a continuous representation Langman et al. (2025), The quantizer then maps this continuous representation to a discrete code from a codebook. Finally, the decoder reconstructs the original waveform from the selected code Agustsson et al. (2017). Compression is achieved when the number of bits used to represent the code is smaller than that required for the original audio signal Yang et al. (2020).

Based on the number of quantizers used, quantization method of codecs can be broadly categorized into two types: multiple stage vector quantization Juang & Gray (1982) and single vector quantization (SVQ). Audio codecs, such as SoundStream Zeghidour et al. (2021), most commonly use residual vector quantizer (RVQ) for quantization. With iterative residual refinement, the multiple stage vector quantizer can decrease loss of information. Multi-codebook codecs depend on multi-sequence prediction, which reduces efficiency and robustness Li et al. (2024). Single vector quantization is simpler and particularly useful for downstream generation tasks such as acoustic language models Ye et al. (2025). Recent work such as WavTokenizer Ye et al. (2024) investigates speech compression using a single codebook. However, existing single quantizer approaches do not take into account more complicated audio signals such as music and environmental sounds.

---

[1] https://anonymous.4open.science/r/Mel_cap_demo-49EF

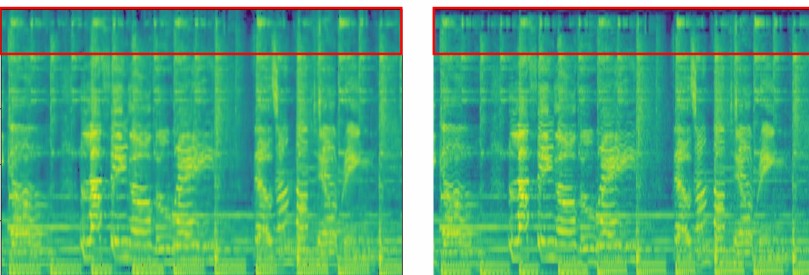

Figure 1: Loss of high-frequency (above 20k Hz) detail in a waveform-based codec. **Left:** spectrogram of result from a waveform-based codec using 4 quantizers. **Right:** ground truth (GT). Noticeable differences exist in the high-frequency Mel spectra, resulting in poor reconstruction of high-frequency components, the bright ringing sound in the original sound.

Neural audio codecs can also be categorized into two types based on the representation they compress: waveform-based and spectral-based approaches. In waveform-based neural codecs, waveforms are directly passed to the encoder. Waveform tokens are typically learned using encoder–decoder architecture trained to reconstruct the waveform Mousavi et al. (2025). EnCodec Défossez et al. (2022) extends this architecture with a multi-scale STFT discriminator, which help reduce artifacts and produces high-quality samples. DAC Kumar et al. (2023) improves this framework by introducing multiscale mel reconstruction loss, which better captures details and thus improves audio quality. SNAC Siuzdak et al. (2024) extends Residual Vector Quantization (RVQ) to multiple temporal resolutions, resulting in more efficient compression.

However, waveform-based approaches still require large model capacity and a greater number of quantizers to capture fine frequency detail accurately seen in Figure 1, which is incompatible with our single-codebook objective. Spectral-based approaches solve this problem by transforming the waveform into the spectral domain, which provides a more effecient representation and allows the model to better capture fine-grained frequency details. Recent works such as APCodec Ai et al. (2024) jointly models amplitude and phase spectra with residual vector quantization and GAN-based training, enabling high-quality 48 kHz audio reconstruction. However, the instability of Generative Adversarial Network (GAN) training hinder the model's capacity to faithfully reconstruct the input audio, particularly subtle frequency details and original phase Wu et al. (2024).

To address the aforementioned challenge, this paper proposed a novel audio codec based on mel-spectrogram, which is a compact representation that can compresses complex audio signals into a single codebook. There are three main contributions of our method. First, this codec incorporates perceptual loss into mel-spectrogram reconstruction to alleviate the over-smoothing problem, and further relates it to the feature matching loss used in traditional GAN-based codecs. Second, we use a two-stage training framework to train Vector Quantized-Variational AutoEncoder (VQ-VAE) and GAN-based Vocoder separately, which leads to better GAN training stability and audio quality. Third, this codec aims to encode high-sampling-rate audio (e.g., 44 kHz) using a single quantizer layer, thus meeting the requirements of downstream generation tasks.

## 2 RELATED WORK

### 2.1 2D TRANSFORMER TOKENIZER

Discrete audio tokenizers often compress audio into a latent one-dimensional representation, which is then quantized into a sequence of discrete tokens. This process is accomplished by compressing the audio along the temporal dimension, so it is defined as a 1D tokenizer. A 2D tokenizer compresses audio in both time and frequency dimensions (typically operating on spectrograms), and then transforms the compressed representation into a sequence of discrete tokens. Many works on 2D tokenizers have been explored in the image domain Yu et al. (2024). VQ-VAE Yang et al. (2020) first introduced vector quantization in the latent space of VAEs to map images, audio, and video into discrete values. Vector Quantized Generative Adversarial Network (VQGAN) Esser et al.

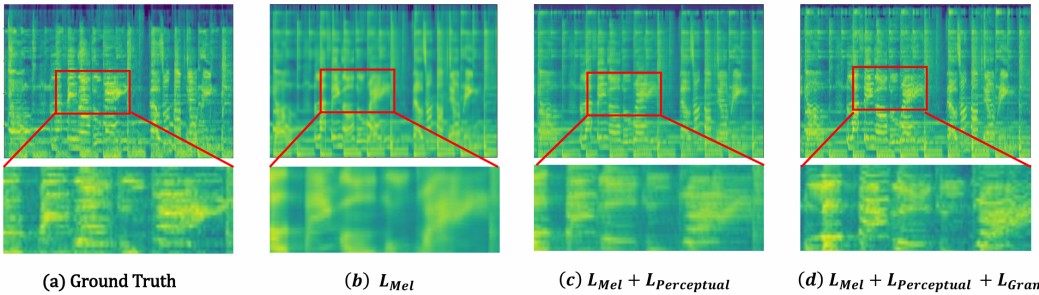

(a) Ground Truth      (b) $L_{Mel}$      (c) $L_{Mel} + L_{Perceptual}$      (d) $L_{Mel} + L_{Perceptual} + L_{Gram}$

Figure 2: Comparison of reconstructed log-mel spectrograms trained with different loss. The bottom row shows a zoomed-in view, highlighting the differences in smoothness and spectral sharpness.

(2021) extended this by adding perceptual and adversarial losses to better capture detail information. ViT-GAN further replaced convolutions with ViT Transformers Dosovitskiy et al. (2021). In this paper, we assume that the latent space of audio should preserve a 2D structure, maintaining an explicit alignment between time and frequency. Building on the structure of powerful 2D Transformer tokenizers NVIDIA et al. (2025), we fully explore compact 2D representations for audio.

## 2.2 VOCODER

Neural vocoders are neural network models that converts intermediate representations, such as mel-spectrograms, into high-fidelity audio Jiao et al. (2021). Autoregressive models had long been the best-performing vocoders. WaveNet van den Oord et al. (2016), for instance, uses the mel-spectrogram as a local condition. However, its requirement for sequential (sample-by-sample) generation limits streaming efficiency. GAN-based models are capable of generating speech from mel-spectrogram efficiently Kong et al. (2020). Since low latency is a key property for a good codec, we build our model on Vocos Siuzdak (2024). Vocos is a fast neural GAN-based vocoder designed to reconstruct audio from mel-spectrogram through inverse Fourier transform.

## 3 METHODS

GAN-based end-to-end codecs require dedicated discriminators and multiple codenooks to improve the waveform details, high-frequency components, and phase synchronization, but training these discriminators is time-consuming and convergence is often slow Wu et al. (2023). Consequently, we propose a two-stage codec, where the first stage focuses on mel-spectrogram reconstruction with metric losses, and the second stage incorporates a discriminator to recover high-fidelity waveform from mel discrete tokens. This architecture significantly improves training efficiency, enabling our second-stage model to converge within only 50 epochs.

### 3.1 LOG-MEL SPECTROGRAM

One efficient way to extract spectral features from an audio signal is through the Short-Time Fourier Transform (STFT). Given an input signal $x[n]$ with length T, $X_t[k]$, the STFT coefficient for the k-th frequency bin and the t-th time frame, denoted as $x_t[k]$. To better connect with the human sound perception, the frequency axis of the spectrogram can be mapped onto the Mel scale using a filter bank. This result is known as Mel spectrogram. Finally, the logarithm of the Mel spectrogram is taken to limit the range of values. The log-Mel spectrogram coefficient for the k-th frequency bin and the t-th time frame is given by:

$$LMS_t[m] = \log \left[ \sum_{k=0}^{N-1} H_m[k] \cdot \left| \sum_{n=0}^{N-1} x[n] \, w[n - tH] \, e^{-j2\pi kn/N} \right|^2 \right]. \tag{1}$$
$$\scriptstyle 0 \le m \le M-1$$

where $H_m[k]$ is the $k^{th}$ coefficient for the $m^{th}$ filter bank 200 (2001), $w[n]$ is the window function (e.g., Hamming window), H is the hop size, and N is the total number of frequency bins.

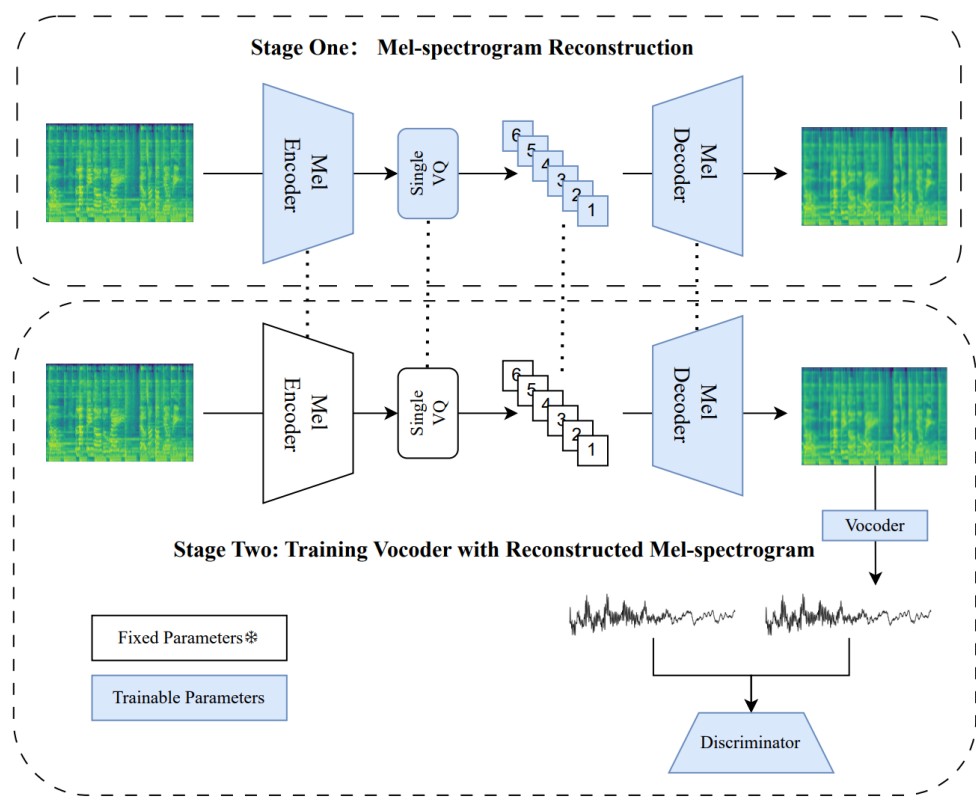

Figure 3: Training paradigm of MelCap.

The total number of filter banks, denoted as M, determines the frequency resolution of the resulting Mel spectrogram. To preserve high-frequency details, the number of Mel filter banks M should be chosen sufficiently large, i.e., not less than 96.

## 3.2 FIRST STAGE: MEL-SPECTRALGRAM RECONSTRUCTION

In the first stage, we compress audio into discrete tokens and then reconstruct the mel-spectrogram from these tokens. For convenience, we adopt the log-mel representation mentioned in equation 1, which helps constrain the value range. Our method builds on the Cosmos tokenizer NVIDIA et al. (2025) as the foundational encoder–decoder. We optimize with the L1 loss applied on the log Mel-spectrograms, which minimizes the element-wise difference between the input and reconstructed spectrograms:

$$\mathcal{L}_{\text{Mel}} = \frac{1}{T \cdot M} \sum_{t=0}^{T-1} \sum_{m=0}^{M-1} \left\| \mathbf{S}_{t,m} - \hat{\mathbf{S}}_{t,m} \right\|_1, \tag{2}$$

where $\mathbf{S}_{t,m}$ and $\hat{\mathbf{S}}_{t,m}$ denote the value of the input and reconstructed Mel-spectrogram, respectively, at the $t$-th time frame and the $m$-th Mel frequency bin. Using only a reconstruction loss can lead to overly smooth reconstructed Mel-spectrograms, as shown in Figure 2. This oversmoothness negatively affects downstream generation tasks such as TTS, causing the synthesized waveform to sound muffled and unnatural Sheng & Pavlovskiy (2018). In order to obtain a more detailed Mel spectrogram, we employ perceptual loss based on the VGG-19 features, given by Simonyan & Zisserman (2015). We provide a theoretical justification for the perceptual loss on mel-spectrograms and the feature matching loss used when training generator of the vocoder in the appendix A.5.

$$\mathcal{L}_{\text{Perceptual}} = \frac{1}{L} \sum_{l=1}^{L} \sum_{t} \alpha_l \left\| \text{VGG}_l(\hat{S}) - \text{VGG}_l(S) \right\|_1, \tag{3}$$

where $\text{VGG}_l(\cdot) \in \mathbb{R}^{T \times M \times C}$ denotes the feature maps extracted from the $l$-th layer of a pre-trained VGG-19 network, $L$ is the number of layers considered, and $\alpha_l$ is the weight assigned to the $l$-th layer. To further enhance fine details, we fine-tune the tokenizer using a Gram-matrix loss Gatys et al. (2016), which emphasizes sharper structures.

$$\mathcal{L}_{\text{Gram}} = \frac{1}{L} \sum_{l=1}^{L} \sum_{t} \alpha_l \left\| \text{GM}_l(\hat{S}) - \text{GM}_l(S) \right\|_1, \tag{4}$$

### 3.3 SECOND STAGE: FROM MEL-SPECTROGRAM TOKENS TO WAVEFORM

Neural vocoders are primarily designed to recover audio waveforms from mel-spectrogram representations Siuzdak (2024). In contrast, our goal is to reconstruct audio waveforms from the mel-spectrogram discrete tokens obtained in the first stage, rather than from the ground-truth mel-spectrograms. Usage of these codes as input to the vocoder has advantage: it help stabilize GAN training in the second stage. Given that the first stage employs a VQ-VAE, the resulting mel-spectrograms contain reconstruction errors. As we have theoretically shown, the mapping from a continuous high-dimensional mel-spectrogram to a finite discrete codebook introduces a upper bound on the propagated error. Because the codes are discrete and belong to a finite codebook, the propagated errors are strictly bounded, preventing extreme deviations and ensuring more robust waveform recovery. By contrast, mel-spectrograms live in a continuous space, where errors cannot be strictly bounded, making the waveform recovery more sensitive to small perturbations.

#### 3.3.1 ANALYSIS: BOUNDED ERROR OF DISCRETE CODES

**Assumption 3.1** (Discrete Code Quantization). Let $\mathbf{s}$ denote the original mel-spectrogram and $\mathbf{c} \in \mathcal{C}$ be the discrete token obtained from a VQ-VAE encoder $E$, where $\mathcal{C}$ is a finite codebook. We assume that the quantization error due to mapping $\mathbf{s}$ to any code $\mathbf{c}$ in codebook is bounded:

$$\|\mathbf{c_n} - \mathbf{s}\| \leq \|\mathbf{c} - \mathbf{s}\| \leq \|\mathbf{c_f} - \mathbf{s}\| = \Delta,$$

where $c_f$ denotes the farthest code, and $c_n$ denotes the nearest code, $\Delta$ depends on the size and codebook dimension.

Let $\mathbf{w}$ denote the waveform reconstructed in the second stage via a neural vocoder $f$. Then the reconstruction is

$$\mathbf{w} = f(\mathbf{c}).$$

**Lemma 3.2** (Lipschitz Bound). *If $f$ is locally Lipschitz continuous with constant $L$, then*

$$\|f(\mathbf{c}_1) - f(\mathbf{c}_2)\| \leq L\|\mathbf{c}_1 - \mathbf{c}_2\|,$$

**Theorem 3.3** (Bounded Waveform Error). *Combining Lemma 3.1 and Lemma 3.2, the error in the reconstructed waveform due to discrete code quantization is bounded:*

$$\|\mathbf{w} - f(\mathbf{s})\| = \|f(\mathbf{c}) - f(\mathbf{s})\| \leq L\|\mathbf{c} - \mathbf{s}\| \leq L\Delta.$$

*Thus, the propagated error from the first-stage discrete token to the final waveform reconstruction is strictly bounded.*

Assuming that the neural vocoder $f$ is $L$-Lipschitz continuous, the error propagated to the waveform $\mathbf{w}$ can be bounded by $L\Delta$. (The vocoder is a composition of convolution network, activation function and ISTFT; a proof that ISTFT is Lipschitz is given in Appendix A.6.)

#### 3.3.2 MODEL ARCHITECTURE

The theoretical analyse provides guidance for vocoder architecture design. For example, in the generator we choose to use the Snake activation function instead of Leaky ReLU. The Snake activation

helps maintain Lipschitz continuity of the vocoder network, as its derivative is bounded by a constant of 1 Ng et al. (2025). Consequently, using Snake can help control the Lipschitz constant $L$ of the vocoder network, limiting the impact of errors from the first-stage mel-spectrogram. Also, in the discriminator we choose to use spectral normalization Miyato et al. (2018) instead of batch normalization, which is designed to guarantee Lipschitz continuity in discriminator.

### 3.3.3 TRAINING OBJECTIVES

Following the Vocos framework, our second-stage training objective consists of two key components: (i) fine-tuning the decoder from the first stage to better align the latent codes with acoustic features, and (ii) training a vocoder that translates mel-spectral codes into time-domain waveforms. To achieve this objective, we employ a combination of loss functions.

**Reconstruction Loss.** Reconstruction loss refers to the L1 distance between the mel-scaled magnitude spectrograms of the ground-truth waveform and the generated waveform Kong et al. (2020). Unlike Yang et al. (2023) that uses 80 mel-spectrogram bins, our setup constrains the number of bins to be no smaller than the mel-spectrogram resolution defined in the first stage, which is 96 mel-spectrogram bins for music and environment sound. This ensures consistency between the first stage and second stage, preventing the loss of high-frequency details when training second stage.

**Feature Matching Loss.** Feature matching loss measures the learned similarity between a real and generated sample via discriminator features Larsen et al. (2016), Kumar et al. (2019) Hifi-GAN Kong et al. (2020) first used it as an additional loss to train the generator of vocoder. In our case, feature matching is used to reduce over-smoothing, serving a similar role as the VGG loss applied in the first stage. Unlike the original setting in Hifi-GAN, where the feature matching loss weight is 2, we increase it to 5.

**Adversarial Loss.** We employ two discriminators—a multi-resolution discriminator (MRD) Kumar et al. (2023) and a multi-period discriminator (MPD)—to enhance perceptual quality via adversarial learning Zeghidour et al. (2021).

## 4 RESULTS

Reconstruct waveform from discrete tokens has become a fundamental task for audio codecs. In this section, we assess the performance of method relative to established baseline codecs.

**DataSets.** The first-version model is trained on the AudioSet dataset, using the entire training subset (bal train). The AudioSet covers a wide range of sounds, including human and animal vocalizations, musical instruments and genres, as well as common everyday environmental noises. We train a second version adding speech dataset hq-conversations. All audio files are kept at their original sampling rate of 44 kHz. For each audio sample, mel-scaled spectrograms are computed with the following parameters: FFT size $n_{fft}$=1024, hop size $hop_n$ =256, and 96 Mel bins.

**Training Details.** In the first stage, we train the mel-spectrogram tokenizer using a combination of L1 reconstruction loss, quantization loss, and perceptual loss until convergence. Afterward, we replace the perceptual loss with a Gram-matrix loss to fine-tune the model, continuing training until convergence. During training, samples are randomly cropped to 24,320 samples, yielding a mel-spectrogram resolution of 96 × 96. We also train a vocoder using ground-truth mel-spectrograms as reference to evaluate the effect of different loss terms. In the second stage, the encoder and quantizer parameters are frozen. We jointly train the tokenizer decoder, vocoder, and discriminator.

**Baseline Methods.** Our proposed model is compared against DAC Kumar et al. (2023), SNAC Siuzdak et al. (2024), Mel Codec Langman et al. (2025), NVIDIA NeMo Audio Codec and WavTokenizer Ji et al. (2025). For all baselines, we use the officially released pretrained checkpoints— the 24kHz version for wavTokenizer and 44 kHz versions for other methods, which are publicly available online.

### 4.1 EVALUATION

We evaluate our models using four primary objective metrics, VISQOL, LSD, Mel Distance, STFT Distance, and two additional reference metrics, UTMOS and V/UV F1. The primary metrics assess

spectral and perceptual fidelity, while the reference metrics are included for informational purposes, as they are designed for speech and may be less reliable in general audio tests.

**VISQOL.** ViSQOL is an objective perceptual audio quality metric that compares reference and degraded audio signals to produce scores correlated with human listening judgments. In this work, we use audio mode, which operates on fullband audio at 48 kHz.

**LSD.** Log-Spectral Distance (LSD) is a widely used objective metric that measures the difference between the log-magnitude spectra of reference and synthesized audio, providing an indication of spectral distortion and overall reconstruction fidelity.

**Mel Distance.** L1 distance between the mel-scaled magnitude spectrograms of the ground truth and the generated sample.

**STFT Distance.** L1 distance between time-frequency representations of the ground truth and the prediction, computed using multiscale Short-Time Fourier Transform (STFT).

**UTMOS.** UTMOS is an automatic mean opinion score (MOS) prediction system that estimates perceptual audio quality and correlates highly with human judgments at sampling rate 16k. However, since we focus on high-frequency details, this 16kHz sampling rate makes UTMOS less suitable for our evaluation.

**V/UV F1.** F1 measures the classification accuracy of voiced and unvoiced segments. Since AudioSet contains diverse sound categories beyond speech, this metric—originally designed for speech—is only indicative in our setting.

## 4.2 ABLATION EXPERIMENT RESULT FOR MEL-SPECTROGRAM RECONSTRUCTION

To investigate the impact of different loss functions used in first stage on the perceptual quality of the generated audio, we conduct an ablation study using a fixed pretrained vocoder. Specifically, we compare three training settings: (1) using only the reconstruction loss, (2) using reconstruction loss combined with VGG loss, and (3) using reconstruction loss, VGG loss, and an additional Gram matrix (GM) loss. This study allows us to analyze how each component contributes to perceptual fidelity.

Table 1: Comparison of different loss terms used in the first stage. MAE denotes the element-wise L1 difference between the input and output mel-spectrograms after the first stage.

| Loss terms | MAE↓ | VISQOL↑ | LSD↓ | STFT Dis ↓ | Mel Dis↓ | F1↑ | UTMOS↑ |
|---|---|---|---|---|---|---|---|
| Reconstruction Loss | **0.26** | **4.36** | 0.67 | **1.68** | **0.48** | 0.58 | 1.31 |
| + VGG loss | 0.31 | 4.26 | **0.64** | **1.68** | 0.50 | **0.63** | 1.31 |
| + GM loss | 0.41 | 4.24 | 0.67 | 1.73 | 0.58 | 0.61 | 1.31 |

Table 1 compares different loss terms used in the first stage. While adding GM loss reduces the over-smoothing of the reconstructed spectrogram, it introduces artifacts, leading to worse overall metrics. Therefore, we choose not to use GM loss in our final model.

## 4.3 AUDIO RECONSTRUCTION

In the second stage, we use the mel-spectrogram tokens obtained from the first stage as input. After training the second stage, our final results are obtained from the jointly optimized decoder and vocoder. Our evaluation, conducted on the AudioSet test set and detailed in 2. The result shows that Mel Cap with VGG loss achieves competitive perceptual quality (VISQOL 4.29) while using only a single codebook. It also obtains the best fidelity metrics (lowest LSD and Mel Distance), outperforming its non-VGG variant, single-codebook baseline and multi-codebook baselines. This demonstrates that incorporating VGG loss effectively mitigates over-smoothing and enhances spectral reconstruction , which is beneficial for training in second stage.

**Subject Evaluation.** We conducted a MUSHRA-style listening test to evaluate the perceptual quality of the generated audio. A total of 15 participants were recruited for the experiment. Each

Table 2: Objective evaluation metrics for different codecs.

| Codec | Codebook Number | Token Rate | Visqol↑ | LSD↓ | STFT Dis↓ | Mel Dis↓ | F1↑ | UTMOS↑ |
|---|---|---|---|---|---|---|---|---|
| DAC | 9 | 774 | **4.46** | 0.67 | 1.77 | 0.65 | **0.87** | 1.31 |
| Mel Codec | 9 | 672 | 4.04 | 0.90 | 2.77 | 0.96 | 0.64 | 1.30 |
| Nvidia Codec | 9 | 672 | 4.05 | 0.89 | 2.81 | 0.96 | 0.82 | 1.30 |
| SNAC | 4 | 240 | 4.35 | 0.68 | **1.69** | 0.68 | 0.76 | 1.31 |
| DAC(s) | 4 | 344 | 4.15 | 0.80 | 2.54 | 0.95 | 0.78 | 1.31 |
| WaveTokenizer | 1 | 75 | 4.20 | 0.71 | 2.30 | 0.76 | 0.65 | 1.31 |
| Mel Cap w/o vgg | 1 | 260 | 4.18 | 0.76 | 1.87 | 0.57 | 0.52 | 1.31 |
| Mel Cap w/ vgg | 1 | 260 | 4.29 | **0.66** | 1.90 | **0.56** | 0.63 | 1.31 |

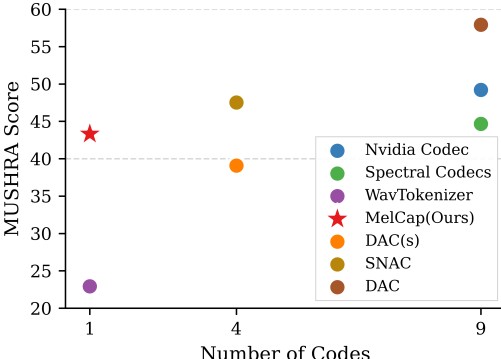

Figure 4: Subjective evaluation metrics calculated for different codecs. Points closer to the top-left indicate that better perceptual quality is achieved using fewer tokens, corresponding to better codec performance.

participant was presented with 30 randomly selected audio samples drawn from a diverse set of test cases including music, speech, and general sounds. For each trial, participants were asked to rate the audio samples on a continuous quality scale, following the MUSHRA protocol. After the test, we aggregated the ratings across all participants and samples to obtain the final statistical results. The subjective evaluation indicates that Mel Cap achieves perceptual quality comparable to other multi-codec approaches.

Table 3: Objective quality on the *Music* dataset. Best/second-best are marked in **bold**/underlined.

| Codec | Codebook Num. | Token Rate | VISQOL↑ | LSD↓ | STFT Dis↓ | Mel Dis↓ | F1↑ | UTMOS↑ |
|---|---|---|---|---|---|---|---|---|
| DAC | 9 | 774 | 4.20 | **0.96** | **2.29** | 0.41 | **0.89** | 1.31 |
| Mel Codec | 9 | 672 | **4.31** | 1.14 | 2.80 | 0.55 | 0.82 | 1.31 |
| Nvidia Codec | 9 | 672 | 3.94 | 1.14 | 2.83 | 0.50 | 0.83 | 1.30 |
| SNAC | 4 | 240 | 4.06 | 1.15 | 3.09 | 0.64 | 0.71 | 1.30 |
| DAC(s) | 4 | 240 | 3.78 | 1.32 | 3.08 | 0.64 | 0.83 | 1.31 |
| WaveTokenizer | 1 | 75 | 2.94 | 1.49 | 3.74 | 0.85 | 0.68 | **1.32** |
| Mel Cap | 1 | 260 | 3.97 | 1.01 | 2.70 | **0.40** | 0.61 | **1.32** |

A key property of a codec is its ability to compress and reconstruct unseen data. After augmenting the training set with hq-conversations Magic Data (2024), we tested the model on unseen music data. The results, as reported in Table 3, demonstrate that our codec generalizes well beyond AudioSet and maintains competitive perceptual and spectral quality in out-of-distribution data.

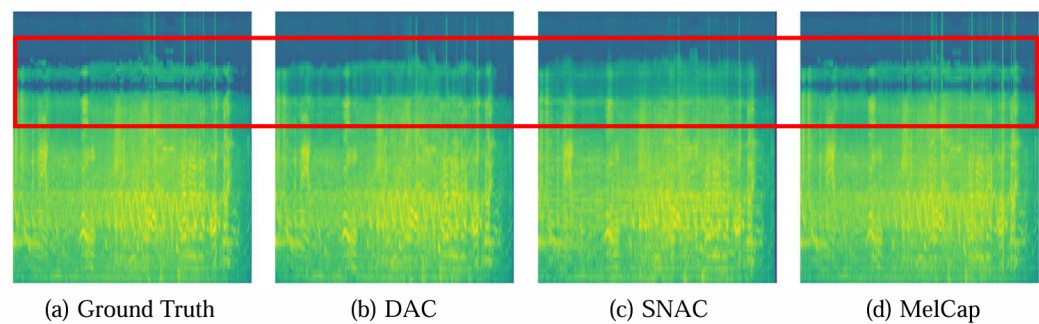

| (a) Ground Truth | (b) DAC | (c) SNAC | (d) MelCap |

Figure 5: Mel-spectrogram comparison of original and reconstructed waveforms produced by different codecs. (a) Ground-truth; (b) DAC; (c) Mel Codec; (d) Nvidia Codec; (e) SNAC; (f) DAC(small); (g) WaveTokenizer; (h) Mel Cap. MelCap accurately reconstructs the high-frequency details.

Table 4: Downstream sound event classification performance. "Reference" refers to results obtained using the ground-truth waveform, while the other columns show the performance using audio reconstructed by different models.

|  | Reference | SNAC | DAC | Our Method |
|---|---|---|---|---|
| **F1**↑ | 0.3899 | 0.3223 | 0.3363 | **0.3398** |
| **mAP**↑ | 0.1626 | 0.1278 | 0.1251 | **0.1345** |

## 4.4 DOWNSTREAM TASK EVALUATION

Unlike speech-only datasets, which can be evaluated using reconstructed waveform quality by ASR models, AudioSet contains multiple sound categories and is designed for audio classification tasks. We further evaluate our codec on the downstream classification task. Specifically, we employ pretrained models from Dinkel et al. (2023) and compute the top-3 predicted labels using the reconstructed waveforms. To assess the codec's ability to preserve semantic information, we report F1 and mAP scores in 4, which measure the accuracy of sound event classification. The performance demonstrates the codec's effectiveness in retaining discriminative detail beyond perceptual quality, achieving better downstream classification results compared to other codec baselines.

## 5 FUTURE WORK

Currently, available open-source high-fidelity audio datasets remain scarce. Even 44kHz or 48kHz corpora often contain upconverted 16kHz content. This limited data quality hinders current compression efforts. In future work, we will collect more high-resolution recordings and continue to improve our model to achieve superior perceptual quality. Given the potential relationship between frequency distribution and model capacity, analyzing how different frequency ranges are encoded may help improve codec efficiency and perceptual quality. More efficient architectures that better preserve frequency details may be achieved by separating signal to multibands. Existing evaluation metrics largely focus on the quality of speech, which may not fully capture the perceptual fidelity of other types of sounds. Therefore, we will focus on identifying or developing suitable benchmarks for evaluating codecs targeted at non-music and non-speech sounds.

## 6 CONCLUSION

To compress complex sounds in natural environments into a single codebook, we conducted a series of explorations. We compressed mel-spectrograms into a single codebook and reconstructed high-quality audio from the discrete mel tokens using a vocoder. During this process, we encountered over-smoothing issues, which we mitigated through careful loss design, and stabilized the training via network architecture improvements. These findings provide new insights for future research in audio codec development.

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

## A APPENDIX

### A.1 USE OF LLMS

We only used large language models (LLMs) as a tool for language refinement and editing. They were not involved in the design of the methodology, experimental setup, data analysis, or any other core aspect of this research.

### A.2 ETHICS STATEMENT

This study has been approved by the relevant ethics committee or institutional review board and was conducted in strict accordance with ethical guidelines. The rights, privacy, and welfare of participants were fully respected and protected, and all personal information was kept confidential.

Informed Consent: All participants were informed of the study's objectives, procedures, potential risks, and benefits, either verbally or in writing, and provided their informed consent.

Data Confidentiality and Privacy Protection: Measures were implemented to safeguard participants' personal information and ensure privacy.

### A.3 REPRODUCIBILITY STATEMENT

Use of Research Data: All research data were collected, stored, and used in accordance with legal and ethical standards, ensuring transparency and proper interpretation.

We ensure that our method is fully reproducible. Upon acceptance of this paper, we will publicly release all code, model weights, and training data necessary to replicate our results.

## A.4 EQUIVALENCE OF SPECTROGRAM VGG LOSS AND VOCODER FEATURE MATCHING LOSS

Let $\hat{x}$ denote the waveform generated by a vocoder, and $x$ the corresponding ground-truth waveform. Define the Short-Time Fourier Transform (STFT) of the waveform as:

$$\hat{S} = \text{STFT}(\hat{x}), \quad S = \text{STFT}(x) \tag{5}$$

### A.4.1 SPECTROGRAM VGG LOSS

A spectrogram-based VGG loss is defined as the L1 distance between feature maps extracted from a convolutional network $\phi$ (e.g., VGG) applied to the spectrograms:

$$\mathcal{L}_{\text{VGG}}(\hat{S}, S) = \sum_{l=1}^{L} w_l \left\| \phi_l(\hat{S}) - \phi_l(S) \right\|_1 \tag{6}$$

where $\phi_l(\cdot)$ is the feature map at the $l$-th layer, $w_l$ is a weighting coefficient, and $L$ is the total number of layers considered.

### A.4.2 VOCODER FEATURE MATCHING LOSS

In vocoder GANs, the feature matching loss is defined using the discriminator $D$:

$$\mathcal{L}_{\text{FM}}(\hat{x}, x) = \sum_{d \in \mathcal{M}} \sum_{l=1}^{L_d} \|D_l^{(d)}(\hat{x}) - D_l^{(d)}(x)\|_1 \tag{7}$$

where $D_l^{(d)}(\cdot)$ denotes the feature map of the $l$-th layer of the $d$-th discriminator, and $\mathcal{D}$ is the set of discriminators (e.g., multi-resolution discriminators).

Each discriminator first computes a spectrogram of the waveform:

$$X = \text{STFT}(\cdot) \tag{8}$$

and then applies a sequence of convolutional layers with non-linearities:

$$D_l^{(d)}(\hat{x}) = \sigma(W_l^{(d)} * X + b_l^{(d)}), \tag{9}$$

where $W_l^{(d)}, b_l^{(d)}$ are the convolutional weights and biases, and $\sigma(\cdot)$ is the activation function.

### A.4.3 EQUIVALENCE

Substituting $X = \text{STFT}(\hat{x})$ and $X = \text{STFT}(x)$ into the feature matching loss, we obtain:

$$\mathcal{L}_{\text{FM}}(\hat{x}, x) = \sum_{d,l} \left\| \sigma(W_l^{(d)} * \text{STFT}(\hat{x}) + b_l^{(d)}) - \sigma(W_l^{(d)} * \text{STFT}(x) + b_l^{(d)}) \right\|_1 \tag{10}$$

Comparing with the spectrogram VGG loss in Eq. (2), we see that the two losses share the same mathematical form:

$$\sum_l \|F_l(\text{STFT}(\hat{x})) - F_l(\text{STFT}(x))\|_1 \tag{11}$$

where $F_l$ denotes a convolutional feature extractor. The only difference lies in the choice of network parameters (pretrained VGG weights vs. learned discriminator weights).

## A.5 EQUIVALENCE OF VGG LOSS AND FEATURE-MATCHING LOSS

Both the spectrogram-based VGG loss and the vocoder feature matching loss are equivalent in the sense that they compute an L1 distance in the convolutional feature space of a spectrogram. Formally,

$$\mathcal{L}_{\text{VGG}}(\hat{S}, S) \approx \mathcal{L}_{\text{FM}}(\hat{x}, x), \tag{12}$$

up to the network weights. This shows that the vocoder feature matching loss can be interpreted as a generalized, learnable spectrogram-based perceptual loss.

## A.6 PROOF OF ISTFT LIPSCHITZ CONTINUITY

We prove that the inverse short-time Fourier transform (ISTFT) operator is Lipschitz continuous with respect to the spectrogram input.

**Lemma A.1** (ISTFT Lipschitz Continuity). *Let $\mathcal{S} \in \mathbb{C}^{F \times T}$ be a complex spectrogram obtained by short-time Fourier transform (STFT) with analysis window $g \in \mathbb{R}^N$ and hop size $H$. Define the ISTFT operator $\mathrm{ISTFT} : \mathbb{C}^{F \times T} \to \mathbb{R}^M$ with synthesis window $h$. Then, for any two spectrograms $\mathcal{S}_1, \mathcal{S}_2$,*

$$\|\mathrm{ISTFT}(\mathcal{S}_1) - \mathrm{ISTFT}(\mathcal{S}_2)\|_2 \ \leq \ L_{\mathrm{ISTFT}} \, \|\mathcal{S}_1 - \mathcal{S}_2\|_2,$$

*where the Lipschitz constant $L_{\mathrm{ISTFT}}$ depends only on the window functions and hop size.*

*Proof.* Recall that ISTFT reconstructs the waveform by overlap-add (OLA) of inverse FFTs of each frame:

$$\hat{x}[n] = \sum_t h[n - tH] \cdot \mathrm{IFFT}(\mathcal{S}[:, t])[n - tH].$$

Let $\Delta \mathcal{S} = \mathcal{S}_1 - \mathcal{S}_2$. Then the waveform difference is

$$\Delta x[n] = \sum_t h[n - tH] \cdot \mathrm{IFFT}(\Delta \mathcal{S}[:, t])[n - tH].$$

By Parseval's theorem, the $\ell_2$ norm of the IFFT is equal to $\ell_2$ norm of the spectrum:

$$\|\mathrm{IFFT}(\Delta \mathcal{S}[:, t])\|_2 = \sqrt{N} \, \|\Delta \mathcal{S}[:, t]\|_2,$$

where $N$ is the FFT length.

Applying Minkowski's inequality to the OLA sum:

$$\|\Delta x\|_2 \leq \sum_t \|h(\cdot - tH)\|_\infty \cdot \|\mathrm{IFFT}(\Delta \mathcal{S}[:, t])\|_2.$$

Since the shifted window has the same maximum magnitude as $h$,

$$\|\Delta x\|_2 \leq \|h\|_\infty \cdot \sqrt{N} \sum_t \|\Delta \mathcal{S}[:, t]\|_2.$$

Finally, by Cauchy–Schwarz,

$$\sum_t \|\Delta \mathcal{S}[:, t]\|_2 \leq \sqrt{T} \, \|\Delta \mathcal{S}\|_2.$$

Combining the inequalities, we obtain

$$\|\mathrm{ISTFT}(\mathcal{S}_1) - \mathrm{ISTFT}(\mathcal{S}_2)\|_2 \ \leq \ \|h\|_\infty \cdot \sqrt{NT} \, \|\mathcal{S}_1 - \mathcal{S}_2\|_2.$$

Thus ISTFT is Lipschitz continuous with constant

$$L_{\mathrm{ISTFT}} \leq \|h\|_\infty \cdot \sqrt{NT}.$$

$\square$

*Remark* A.2. In practice, when $h$ is chosen as the canonical synthesis window satisfying the perfect-reconstruction condition (e.g., Hann window with 50% overlap), $\|h\|_\infty \leq 1$. Hence the ISTFT operator has a moderate Lipschitz constant that scales with the FFT length $N$ and number of frames $T$, ensuring stability against spectrogram perturbations.

