# OpenReview forum: "MelCap: A Unified Single-Codebook Neural Codec for High-Fidelity Audio Compression"
_ICLR.cc/2026/Conference — ICLR 2026 Conference Withdrawn Submission_

### Official Review · Reviewer_amqW · 2025-10-29

**Soundness:** 2
**Presentation:** 2
**Contribution:** 2
**Rating:** 2
**Confidence:** 4

**Summary:**

The paper proposes MelCap, a two-stage neural audio codec that compresses log-mel spectrograms into a single codebook using a 2D tokenizer, followed by a GAN-based vocoder. A perceptual VGG loss is added to improve mel reconstruction. Experiments include objective metrics, a MUSHRA-style listening test, and a small downstream comparison based on reconstructed waveforms.

**Strengths:**

* The two-stage, single-codebook formulation is simple and practically appealing.
* Objective metrics appear competitive against several recent codecs.
* Perceptual losses in stage 1 are ablated and seem beneficial.
* The overall training pipeline is conceptually clear and could be attractive for downstream modeling.

**Weaknesses:**

* The paper does not provide the actual bitrate of MelCap. Token rate, number of codebooks, or relative code lengths are not proxies for bitrate. Without bitrate normalization, the comparisons are not meaningfully interpretable, and claims of competitiveness are not supported.
Figure 4 plots performance against “number of codebooks”. Different codecs operate at different tokenization rates, and some baselines do not even have a consistent frame/token rate across codebooks. The perceptual results should be plotted against bitrate, not number of codebooks. Additionally no confidence intervals are shown and the score of the reference signal is missing.

* To evaluate usefulness for modeling, one expects experiments directly on the discrete tokens (e.g., token-based ASR/TTS, audio LMs). The paper evaluates waveform reconstructions using pretrained classifiers, which mostly measures vocoder fidelity, not the utility of the tokens for downstream modeling tasks.

* The method claims to use a previously proposed tokenizer architecture but offers very few implementation details. Critical parameters (depth, channel counts, quantizer settings, receptive field, training schedules) are missing. Several important discriminator and vocoder design details are also unspecified.

* The theory lacks proper rigour: The section starts by comparing discrete tokens c with mel-spectrograms s, which is problematic. It then continues to show that the vocoder f is Lipschitz-continuous, but f operates on a discrete domain. With proper definitions and analysis one could fix the theorem, but the result is also not surprising as the theory is only used to motivate the use of snake activations (which is already commonplace) and spectral normalization. Ablations for both decisions are missing.

* Stage 1 trains a mel decoder, but stage 2 fine-tunes that decoder together with the vocoder. It is unclear:
    * why the decoder isn’t trained directly on token distributions from the start,
    * whether fine-tuning is required,
    * what happens if the decoder is frozen.
No ablations compare these options.
* The paper devotes substantial space to preliminaries (mel definitions, metrics) while providing only superficial descriptions of core components, leaving readers unable to understand where improvements come from.
* The abstract claims real-time decoding, but the paper does not report real-time factors, latency, memory consumption, or comparisons against baselines on the same hardware.

**Questions:**

* What bitrate (kbps) does MelCap operate at for each reported operating point?
* Can you evaluate token usefulness directly rather than via reconstructed waveforms?
* Can you provide full architecture tables (layers, channels, strides, codebook sizes, quantization details)?
* Can you support the real-time claim using RTF and latency data?
* Can you formalize the theoretical section: define domains, norms, and mappings?

---

### Official Review · Reviewer_XUpc · 2025-10-31

**Soundness:** 1
**Presentation:** 1
**Contribution:** 1
**Rating:** 2
**Confidence:** 4

**Summary:**

The authors propose MelCap, a single-codebook neural audio codec designed to handle diverse audio sources. MelCap adopts a two-stage training paradigm. In the first stage, a single codebook is trained on Mel-spectrogram reconstruction using a perceptual loss in addition to Mel loss to mitigate reconstruction artifacts. In the second stage, a vocoder is trained to recover waveforms conditioned on the discretized Mel tokens. Objective and subjective evaluations show that MelCap achieves performance comparable to multi-codebook codecs.

**Strengths:**

1. The motivation to unify compression for general audio within a single-codebook framework is reasonable and timely.

**Weaknesses:**

1. The two-stage training paradigm is not novel. Similar approaches have already been proposed in prior work [1] for vocoder-based audio reconstruction.
2. The experimental setup is unconvincing. MelCap is trained specifically on the AudioSet dataset, whereas the pretrained baselines were trained on different sources. Furthermore, the comparison is made across models with different token rates, making the results not directly comparable. The authors should reproduce at least some baselines under consistent conditions and present results at matched token rates.
3. The proposed VGG perceptual loss does not consistently improve performance. In Table 1, MAE, Mel Distance, and VISQOL actually worsen when the VGG loss is applied.
4. The error bound claim for discrete codes is confusing and lacks rigor.
5. The paper suffers from numerous editorial and presentation issues, including mismatched notation (e.g., line 248), incomplete figure given the captions (e.g., Figure 5), incorrect inline citations, and multiple grammar and formatting problems.

[1] Low Bit-Rate Speech Coding with VQ-VAE and a WaveNet Decoder, Gârbacea et al., 2019

**Questions:**

1. In Section 4.1, the authors state that UTMOS and V/UV F1 metrics are unsuitable for evaluation. Why are these metrics still reported in the results?
2. In the proof of the error bound, why does the decoder function $f$ takes a single code as input? To recover the full waveform $f$ should not be a univariate function, and the bound assumption on the quantization error is not reasonable.

---

### Official Review · Reviewer_pUMk · 2025-11-01

**Soundness:** 2
**Presentation:** 1
**Contribution:** 1
**Rating:** 0
**Confidence:** 5

**Summary:**

The paper present a 2d Mel tokenization-based neural audio codec with 2 stage training from Mel-spectrogram reconstruction to waveform generation.

**Strengths:**

The paper introduces a 2D tokenizer for a neural audio codec. While the idea of compressing audio in a manner similar to image compression is interesting, the paper does not demonstrate the effectiveness of the proposed method. It would be beneficial to include an ablation study using 1D tokenization within the proposed two-stage training framework, and to compare the performance with respect to tokens per second.

**Weaknesses:**

My primary concern lies in the lack of novelty and limited evaluation.

[Effectiveness of 2d Tokenization]

The paper does not verify the effectiveness of the proposed 2D audio tokenization. It is difficult to understand the rationale for adopting 2D tokenization in a neural audio codec, apart from frequency compression. For neural audio codecs, it is essential to demonstrate the model’s effectiveness in downstream generative tasks, such as audio generation or text-to-speech synthesis.

However, the paper only conducts sound event classification experiments, and the baselines used for comparison are highly questionable. Moreover, the experimental details are insufficiently described.

[Model]

I cannot find any clear novel contribution in the model design, including the loss functions used for Mel-spectrogram reconstruction or waveform reconstruction via GANs.

[Evaluation]

It is very difficult to assess the model’s performance due to limited evaluation and the use of inappropriate metrics.

To avoid confusion, it would be better to separate the test set into audio, speech, and music subsets.

Additionally, UTMOS is a metric specifically designed for speech quality evaluation. Please refrain from overusing this metric, and apply it only to the speech subset. I also urge the authors to clarify why this metric is used for your model. Do you truly understand the goal of your task and your evaluation metrics?

[Audio Quality]

The audio quality presented in the demo page and in Figure 4 is poor.

Furthermore, the number of test samples is too small to allow any meaningful generalization of the evaluation.

[Downstream Generative Task]

As stated in line 94, the paper claims the potential for high-sampling-rate audio generation. However, the paper does not describe how to decode 2D tokens for downstream generative tasks.

Currently, many works explore multi-token prediction for RVQ token reconstruction, but this paper provides no explanation or method for 2D token prediction.

In summary, the only contribution appears to be the introduction of 2D tokenization for neural audio codecs. However, the current manuscript fails to demonstrate its effectiveness and lacks sufficient empirical validation.

**Questions:**

.

**Details Of Ethics Concerns:**

.

---

### Official Review · Reviewer_GD5i · 2025-11-03

**Soundness:** 2
**Presentation:** 2
**Contribution:** 2
**Rating:** 2
**Confidence:** 5

**Summary:**

This paper addresses the problem of neural audio coding and introduces MelCap, a high-fidelity codec that utilizes a single codebook. MelCap consists of two distinct stages. In the first stage, a mel-spectrogram is extracted from the audio signal and quantized using a 2D tokenizer. In the second stage, a neural vocoder is trained to reconstruct the phase and convert the representation back into a time-domain signal. Both objective and subjective evaluations indicate that MelCap achieves quality comparable to state-of-the-art multi-codebook codecs.

**Strengths:**

1.	The proposed method appears relatively simple and straightforward.
2.	The reported results indicate that the method outperforms the evaluated baselines.
3.	The authors provide some theoretical justification to motivate their model.

**Weaknesses:**

1.	The novelty and overall contribution of the proposed method appear limited relative to prior work.
2.	The experimental results are not fully convincing. In certain cases, there are inconsistencies between the reported figures in this manuscript and those presented in the original papers (e.g., the WavTokenizer subjective evaluation results).
3.	Some design choices in the proposed approach, such as the use of a 2D tokenizer, appear unconventional and unexplained from a signal processing perspective.
4.	The paper’s presentation requires improvement — several figures are not properly referenced, and citations are inconsistently or incorrectly applied.

**Questions:**

1.	The authors propose using a 2D tokenizer over the mel-spectrogram, effectively treating it as a raw image. However, since each column of the mel-spectrogram represents the frequency spectrum of a specific time window (and is therefore highly correlated, often containing harmonics and overtones), it is unclear why a 2D tokenizer was chosen instead of the standard 1D tokenizer commonly used in speech processing.
2.	The paper lacks a discussion of limitations. In particular, latency and lookahead time—especially given the mel-spectrogram configuration—are not addressed. The authors should include a discussion on this aspect.
3.	The description of the second stage of the proposed method is unclear. Are the decoder and vocoder optimized jointly? If so, is the optimization based solely on the vocoder loss, or does it also include a mel-spectrogram reconstruction loss? This should be clarified, as the explanation appears to rely primarily on Figure 3.
4.	Equation (4) is introduced as part of the proposed method, yet the first experiment demonstrates that it does not provide any improvement. The authors should clarify why it was included as a core component of the approach.
5.	Citations are inconsistently applied, and several figures are not referenced within the text.
6.	The reported MUSHRA scores differ substantially from those presented in the WavTokenizer paper. The manuscript reports scores around 20–25, whereas the original paper reports scores above 90. While subjective evaluations may vary, such a large discrepancy raises concerns. The authors should verify whether the correct models were used and whether they were implemented appropriately.
7. Sample page does not work.

---

### Note · Authors · 2025-11-12

I have read and agree with the venue's withdrawal policy on behalf of myself and my co-authors.